# Newborn Screening by Genomic Sequencing: Opportunities and Challenges

**DOI:** 10.3390/ijns8030040

**Published:** 2022-07-15

**Authors:** David Bick, Arzoo Ahmed, Dasha Deen, Alessandra Ferlini, Nicolas Garnier, Dalia Kasperaviciute, Mathilde Leblond, Amanda Pichini, Augusto Rendon, Aditi Satija, Alice Tuff-Lacey, Richard H. Scott

**Affiliations:** 1Genomics England Ltd., Dawson Hall, Charterhouse Square, Barbican, London EC1M 6BQ, UK; arzoo.ahmed@genomicsengland.co.uk (A.A.); dasha.deen@genomicsengland.co.uk (D.D.); dalia.kasperaviciute@genomicsengland.co.uk (D.K.); mathilde.leblond@genomicsengland.co.uk (M.L.); amanda.pichini@genomicsengland.co.uk (A.P.); augusto.rendon@genomicsengland.co.uk (A.R.); aditi.satija@genomicsengland.co.uk (A.S.); alice.tuff-lacey@genomicsengland.co.uk (A.T.-L.); richard.scott@genomicsengland.co.uk (R.H.S.); 2Medical Genetics Unit, Department of Medical Sciences, University of Ferrara, 44121 Ferrara, Italy; fla@unife.it; 3Pfizer Inc., Collegeville, PA 19426, USA; nicolas.garnier@pfizer.com

**Keywords:** newborn, screening, genome, sequencing

## Abstract

Newborn screening for treatable disorders is one of the great public health success stories of the twentieth century worldwide. This commentary examines the potential use of a new technology, next generation sequencing, in newborn screening through the lens of the Wilson and Jungner criteria. Each of the ten criteria are examined to show how they might be applied by programmes using genomic sequencing as a screening tool. While there are obvious advantages to a method that can examine all disease-causing genes in a single assay at an ever-diminishing cost, implementation of genomic sequencing at scale presents numerous challenges, some which are intrinsic to screening for rare disease and some specifically linked to genomics-led screening. In addition to questions specific to routine screening considerations, the ethical, communication, data management, legal, and social implications of genomic screening programmes require consideration.

## 1. Introduction

### 1.1. Newborn Screening Practice Today

For more than 60 years, biochemical screening for treatable disorders in the newborn has proven effective in preventing or dramatically ameliorating the adverse consequences of these conditions [1]. Screening programmes can be found in countries worldwide. The number of disorders on each screening panel varies by country and within countries [2]. Since screening was first deployed in 1961 for phenylketonuria, additional disorders have been added. New technologies such as tandem mass spectrometry (MSMS) have allowed newborn screening programmes to add more conditions at lower cost and more efficiently [1]. By 2015, the United States Secretary of Health and Human Services’ Advisory Committee on Heritable Disorders in Newborns and Children recommended the inclusion of 32 conditions on its Recommended Uniform Screening Panel. These have been adopted by all states, with some state-to-state variability [2]. At present, disease-specific analytes are added individually, limiting the range of rare diseases that can be approached. Further, the current process of ‘adding’ to the existing list of disorders may at some point in the future be constrained by the amount of blood on the newborn screening blood spot card.

Efforts to apply genetic methods to newborn screening using next generation sequencing (NGS) have been underway for several years [3,4,5,6,7,8,9,10], but we are entering a new phase where a number of large studies are launching around the world to generate evidence on whether and how genomics can be used in screening [11,12,13].

In this paper we review some of the opportunities, challenges, and potential approaches to the use of next generation sequencing in newborn screening. We focus primarily on issues for consideration but provide some suggestions on how conditions might be chosen and how screening by genomic testing could be implemented.

### 1.2. Genomic Technology in Newborn Screening

As occurred with MSMS, next generation sequencing (NGS) offers the promise of screening for more disorders at lower overall cost per disease. Screening by NGS is carried out through gene panels, whole exome sequencing (WES), or whole genome sequencing (WGS) [14]. WES and WGS together are referred to here as genome sequencing (GS).

A targeted gene panel can be used for newborn screening [15,16]. Targeted gene panels can be designed “on demand” to enrich the sequencing library with specific (targeted) gene regions to be sequenced. High coverage and in-depth reading, typical of small and medium output gene panels, can identify small nucleotide variants (SNVs) and small indels as well as copy number variants (CNVs) overlapping the gene panel of interest.

WES is designed to explore the protein coding regions and adjacent intronic regulatory sequences of the nuclear genome, which makes up about 2% of the genome. This is where most of the disease-causing variants have been identified. WES is typically performed by first ‘capturing’ the exon-containing DNA from a clinical sample followed by next-generation sequencing (NGS) of that DNA. WES identifies SNVs and indels up to 50 base-pairs (bp) with very high accuracy but fails to identify some small copy number variants (CNVs) as well as structural variants [17]. In contrast WGS sets out to examine the entire DNA sequence of the genome, both the coding and non-coding nuclear sequence as well as the mitochondrial sequence. WGS can evaluate SNVs, indels, non-coding regions, mitochondrial variants, all sizes of CNVs and structural rearrangements. 

In clinical practice today, gene panels, WES and WGS primarily use ‘short-read’ NGS technology. Short-read NGS produces millions of short lengths of DNA sequence, generally 150 base pairs to 300 base pairs in length. Recently, ‘long-read’ NGS sequencing has become available with read lengths of 10,000 base pairs and longer that greatly improves detection of CNVs, structural variants, and variations in complex regions [18]. 

The raw sequencing data produced by NGS is processed by automated bioinformatics pipelines that identify variants by first mapping the reads to a reference genome and then identifying positions that differ from the reference sequence, a process referred to as secondary analysis. Secondary analysis of WGS data typically identifies 3–5 million variants per individual, while WES results in a few hundred thousand variants and panels still fewer. Panels and WES are less expensive than WGS; however, WGS provides a more complete look at the genome [17]. For example, WGS covers a higher proportion of the exome than WES and is therefore more thorough. In addition, WGS can better detect certain clinically relevant variant types such as small duplications and deletions than WES. Detection of non-coding variants, an increasingly important source of pathology in monogenic disorders [19], requires WGS. Some genes such as SMN1 and SMN2 reside in a chromosomal region with many complicated segmental and inverted segmental duplications and therefore require purpose-built informatics pipelines to detect pathogenic variants even when WGS is used [20].

The variants identified by GS are subject to tertiary analysis wherein each variant in each gene is annotated and assessed for evidence of pathogenicity using accepted criteria [21,22,23]. Nearly all variants found through GS are benign or likely benign based on their consequence to protein or frequency in the population and can, therefore, be removed from further consideration with automated filtering algorithms. When GS is used as a diagnostic test, the tertiary analysis pipeline cannot be entirely automated due to the complexity of genotype–phenotype relationships and the difficulties handling this computationally [24]. In the setting of newborn screening, however, it may be possible to automate virtually the entire process for preselected variants and certain types of variants such as frameshift, nonsense, and canonical splice variants in genes where loss of function is the mechanism of disease.

### 1.3. Choosing Conditions, Genes, and Variants for Screening

At the outset of the process of choosing conditions and their associated genes it must be recognized that the cost of genome sequencing and automated analysis for 10 genes or 500 genes is nearly the same once the bioinformatics pipeline is established. Furthermore, there is little additional cost to adding genes to this part of the screening process. That said, the cost of care for the additional conditions in asymptomatic newborns versus the cost in symptomatic patients identified later needs to be kept in mind.

In a newborn screen for treatable disorders of childhood, only a small proportion will screen positive. One available frequency estimate of 1 in 182 newborns for 394 childhood-onset treatable disorders [25,26] suggests that a screen of 100,000 newborns will find only 549 infants with treatable conditions. This is an underestimate as the site includes an additional 307 childhood-onset treatable disorders without a published frequency estimate. 

The scalability of NGS offers an unprecedented opportunity to address a major global health issue: the diagnosis and management of rare diseases, including rare hereditary cancers. About 80% of rare diseases have a genetic origin, and most rare diseases manifest during childhood [17]. This is one of the main rationales behind the creation of the Screen4Care consortium, a European collaborative Innovative Medicines Initiative [13] and the Newborn Genomes Programme in the United Kingdom [12]. The criteria for choosing conditions discussed in this paper are examined for their relevance and applicability to screening for rare diseases using a molecular genetics approach. 

Broadly speaking, there are two DNA sequencing approaches available to assess genes and find the relevant variants for newborn screening: a purpose-built narrowly targeted gene panel and GS where only a targeted set of genes are analysed. Both proceed with the assumption that delivering complete GS information to families would not be actionable and likely unacceptable to most families. 

Wilson and Jungner [27] set out principles for choosing conditions that can be used for screening and reporting (Table 1). Examining these principles can help create a set of criteria used to prioritizing genes for newborn screening. Updated principles have been proposed [28] and a systematic review of principles to be applied in genomic newborn screening was recently published [29]. In the United Kingdom, the National Screening Committee uses guidelines that follow the Wilson and Jungner criteria [30]. Many of the issues raised by GS as a screening test are the same as those present in current newborn screening tests, yet there are some which are unique to GS. Similarities and differences between current newborn screening and GS screening are highlighted.

## 2. Choosing Conditions in Light of Wilson and Jungner Principles

### 2.1. The Condition Sought Should Be an Important Health Problem

Consistent with current screening practice, a newborn sequencing programme could choose genes associated with treatable disorders that appear in childhood and where there will be significant morbidity and mortality if left untreated [26]. This criterion is not at odds with the severity (acute or chronic) of rare diseases in general. Other published criteria exist [4,31]. The concept of what constitutes a significant health problem differs based on individual and family perceptions [32].

Whole genome sequencing may also present the opportunity to look for treatable infectious disorders such as congenital cytomegalovirus virus (CMV) infection; 10–15% of apparently asymptomatic infants with CMV are found to have hearing loss at birth or have delayed-onset loss [33].

Newborn screening for a disorder that is, initially, of more value to the parents than the child is another criterion that some suggest could be considered. The genes associated with hereditary breast and ovarian cancer are good examples. While this may help a parent it will not directly benefit the child’s health and conflicts with the autonomy of the child to decide whether they want to learn about adult-onset conditions. Many in the public and experts would not favour reporting disorders that are primarily of value to parents [34].

### 2.2. The Natural History of the Condition, including Development from Latent to Declared Disease, Should Be Adequately Understood

While there are several genetic disorders that are sufficiently common to provide a guide to the natural history of the condition, many rarer disorders are known that manifest as severe disease but where the full range of the phenotype is unknown. This points to a longstanding challenge faced by the current newborn screening programmes. When a condition is rare there is too little information to be certain that screening is worthwhile. But without extensive data from a screening programme for the specific disease the information needed to add the disease cannot be gathered. The cost of adding a specific analyte for a particular disease to a newborn programme is high, so there must be a great deal of evidence before programmes undertake a pilot to add a new disease.

Screening by genome sequencing may solve the cost problem when adding a disease to a screen but other challenges remain, most notably around penetrance and expressivity. Current Pompe disease (PD) screening, while not a molecular screen, highlights this issue. When screening for PD was first considered for addition to newborn screening it was not clear how many children with a positive screen would turn out to have an adult-onset form [35]. Once screening was underway, it was found that most PD patients had a late onset form of the disease that did not require treatment in the neonatal period. 

In much the same way, the age of onset and severity of many rare but treatable genetic disorders are unknown before a screen for those disorders is initiated. It is therefore important to let families know that, at the beginning of a research programme utilising GS, the screen will include conditions with more or less evidence for inclusion in the panel of genes tested. This is the only way to effectively determine which conditions should continue to be sought in the long term. This also provides opportunities to detect rare conditions that are more prevalent in specific minority ethnic groups. If sequencing becomes part of standard newborn screening, there will need to be a process for adding conditions. This ongoing research will be made easier by the fact that the underlying process of GS and reporting will be in place.

It is useful to know the incidence in the population of symptomatic cases for each disease on the GS panel, where possible. This can help assess whether the screen is finding the expected number of affected newborns. In the United Kingdom, for example, this is regularly assessed for the current newborn screen [36].

### 2.3. There Should Be a Recognizable Latent or Early Symptomatic Stage

There are many treatable genetic disorders that do not present near the time of birth but require months or years to manifest [27]. This is a key attribute of any disorder under consideration if there is to be a timely intervention. Some, such as ornithine transcarbamylase deficiency, usually present before a screening test yields a result but could, nevertheless, be worth including as some affected individuals present after the newborn period [37]. Others, such as familial hypercholesterolemia (FH) will not manifest with coronary artery disease before the infant reaches their 30s. Yet, some current guidelines propose dietary management starting in early childhood and statin therapy starting at age 8 [38].

### 2.4. There Should Be a Suitable Test or Examination 

Like all screening tests, it is important to consider sensitivity, specificity, and positive predictive value when choosing the diseases, genes, and variants in the screening panel. The conditions that will be included in GS screening panels are all rare conditions and therefore the specificity of the variants chosen must be high to achieve an acceptable positive predictive value in the setting of an apparently asymptomatic newborn [39]. 

In GS, when there is strong evidence that a variant or class of variant is disease causing and absent from individuals without the disease, it will have higher specificity. The need to carefully select variants with high specificity is counterbalanced by the adverse impact that it will have on sensitivity. This balance between sensitivity and specificity differs between genes. For example, for the genes associated with cystic fibrosis (CFTR) and spinal muscular atrophy (SMN1) where a large fraction of the causal variants is known, GS will have high sensitivity. This is the ideal situation when screening for rare genetic diseases by GS; the test finds most of the affected newborns with very few false positives. 

Unfortunately, for many genetic disorders there are only a limited number of cases and therefore very few known pathogenic variants. Even if these known variants are combined with presumed pathogenic variants (such as premature stop variants and frameshift variants in disorders where loss of function is the mechanism of disease), the sensitivity is lower. 

Emphasising this need for caution, a recent study suggests that selecting variants that are *not known* to be pathogenic or likely pathogenic (i.e., variants with reduced specificity) results in an increase in the selection of false positive variants. In this study of a clinically unselected population of individuals from the UK Biobank for rare non-synonymous variants in monogenic disease genes the authors note that in large gene panels (>500 genes) nearly all individuals tested have at least one rare non-synonymous candidate diagnostic variant. These are likely to be false positives [40].

### 2.5. The Test Should Be Acceptable to the Population

The current newborn screen uses a blood spot card sampled from a heel prick. It is carried out with one or two samples [41,42]. Sampling beyond 24 h is generally required for optimal screening performance [43]. This has been found acceptable in newborn screening programmes world-wide. In the latest report from The National Health Service (NHS) newborn blood spot screening programme in the United Kingdom, for example, 97.9% of babies were tested and recorded on the Child Health Information System (CHIS) at 17 days [36]. The blood spot card is obtained on day 5 of life in the United Kingdom.

To provide the maximum benefit, the collection of the sample for genomic testing should occur as soon as possible after birth. Umbilical cord blood likely represents a suitable source. In most instances it is readily available in quantities more than sufficient for DNA sequencing. The logistics of collection are challenging in complicated births and in ‘out-of-hospital’ births. A blood spot card sampled from a heel stick or saliva from a cheek swab may also be suitable sources of DNA. Further study is needed to show which is most suitable. 

Programs will need to assess acceptability of longer-term storage of genomic data after screening to support, for example, improvements to screening and new discovery. Acceptability will likely vary between countries and may change over time.

All steps of a proposed screening program must be acceptable to families. To enable sample collection at birth or soon thereafter, recruitment for participation in a GS newborn screen should take place during pregnancy, ideally in the third trimester when health care providers are discussing delivery. Systems such as online consent modules would allow for sufficient education and opportunity to ask questions [44]. It is important to appreciate that knowledge of newborn screening may be incomplete, with parents often not aware of certain aspects of screening [45]. In one study, 59.1% of mothers were unsure or did not know what newborn bloodspots are in the context of newborn screening [46]. Education of couples and perhaps newborn screening preference studies will be needed.

Despite the great potential and flexibility of sequence-based testing, it is unlikely to replace current newborn screening. Many cases of congenital hypothyroidism do not have an identifiable molecular basis. Additionally, the analyte-based screen now in use has greater sensitivity and specificity for the disorders currently tested than genomic testing [8]. Every newborn may therefore require two samplings to optimize the value of both GS and the existing newborn screen. Since GS will likely be an adjunct to the current screen, programmes will need to coordinate GS results with current screening results for disorders evaluated by both genome testing and the existing newborn screen. Careful consideration of discrepant results will be required.

Some disorders identified by GS have additional medical consequences to the child and family. If a hemizygous variant for an X-linked disorder is found in a female newborn this raises the possibility of Turner syndrome. The parents of a newborn found to have Fanconi syndrome due to biallelic variants in BRCA2 are at increased risk for cancer [47]. Parents should be made aware of these possibilities.

The speed of reporting a screening result is an element of acceptability. A screen that does not permit timely intervention will not be accepted. At present, many screening programmes aim to report within 7 days of birth [41]. A consideration of the GS timeline from sampling to family contact with results suggests that this will take longer than a week (Table 2). For some metabolic disorders the patient is symptomatic before the GS screen can be completed [48]. It is, therefore, important to reduce the time taken to report a screening result. Additionally, like current screening programmes, a GS programme will need to keep track of the infant between sampling and reporting. While most will be at home at the time of reporting, some will be in a hospital due to a condition on the newborn screen or other reasons.

### 2.6. There Should Be an Agreed Policy on Whom to Treat as Patients

For each disease in current newborn screening panels, a ‘case definition’ is developed. A case definition in this context is a set of standard criteria used to classify whether an infant has a particular genetic disorder and should therefore be treated. Efforts to create case definitions for newborn screening are underway [49,50], along with work to define terms such as ‘screen positive’ and ‘screen negative’ [51]. Screening for cystic fibrosis [52] and severe combined immunodeficiency [53] demonstrate both the challenges and importance of developing case definitions. Developing a case definition for each condition sought in a GS screening programme will be useful in providing families with uniform care and assessing a programme’s success. 

For certain treatable genetic disorders, molecular testing is the only method to confirm a suspected diagnosis in a symptomatic patient and the only method to diagnose an asymptomatic newborn. Spinal muscular atrophy is an example [54]. To avoid unnecessary treatment given the other limitations regarding knowledge of natural history and penetrance described above, screening asymptomatic newborns for disorders where there *are* non-molecular confirmatory tests is preferred. As is the practice with current screening technologies, the use of a diagnostic test with a high positive predictive value in screen-positive infants gives both physicians and families confidence to move forward with further monitoring and treatment when the GS test is positive.

### 2.7. There Should Be an Accepted Treatment for Patients with Recognized Disease

Similar to the current screening panel, each disorder on the GS screening panel should have a treatment that improves the child’s quality of life and that is readily available to the family. Factors including the risks associated with the treatment, the percentage of cases treated successfully, the degree of improvement, and risks of unnecessary or overtreatment must be considered as well as whether presymptomatic treatment is superior to treatment after symptoms arise. For example, severe combined immunodeficiency, often requires a potentially dangerous procedure, hematopoietic stem cell transplant [53], while xeroderma pigmentosum is managed by avoiding ultraviolet radiation [55].

GS screening programmes will need to contemplate access to other treatments such as off-label use of medications [56] and experimental treatments [57] and whether to include those genes where an experimental treatment or an established treatment is only available in another country. Some parents might choose an existing experimental therapy where the chance of success is uncertain while others would not. 

Some, for example EURORDIS, argue that it is broader “actionability” rather than simply “treatability” that should be sought in the disorders chosen [58]. This broader “actionability” might include conditions where interventions beyond those that are typically considered as treatments might bring benefits, for example, early surveillance for potential comorbidities or facilitated access to social care support. The potential for screening to inform understanding of risk for couples in future pregnancies and to inform reproductive options are also highlighted. There is a diversity of views on this topic. Some point to the benefits of including genes that are actionable even if the condition is not treatable. Others highlight the parents’ loss of the “golden years” when they did not know that their child had an untreatable condition [59,60]. Preference studies are needed to assess the position of parents with respect to these issues.

### 2.8. Facilities for Diagnosis and Treatment Should Be Available

When additional disorders are proposed for newborn screening using the existing technology, care is taken to ensure that diagnostic tests, access to treatment and the necessary healthcare workforce are in place. A pilot programme of GS that may include several hundred disorders will need a care pathway for each condition involving many different paediatric specialties and appropriate regulatory approval. 

Resource issues need to be explored prior to the roll-out of any GS newborn programme. As the disorders are individually rare, this mitigates the impact on workload of laboratory and clinical teams. Clearly, the impact will depend on how many conditions are screened and the thresholds used for return of positive findings, noting that children with a true positive GS screen (those destined to have symptomatic disease) *already* have the disease and will eventually come to specialist attention. 

Engaging with specialists for each disorder who will contact families of screen positive newborns and rapidly initiate confirmatory testing and treatment is essential. Ideally, a ‘parental communication checklist’ for each disorder would be useful during the initial contact with families [61]. Long-term support for families also needs to be available.

### 2.9. The Cost of Case-Finding (Including Diagnosis and Treatment of Patients Diagnosed) Should Be Economically Balanced in Relation to Possible Expenditure on Medical Care as a Whole

Economic modelling is needed to assess the value of screening tests in genetics [62] and newborn screening for genetic disorders [63,64]. Because GS newborn screens involve hundreds of conditions, it may be difficult to develop robust economic models. An evaluation of WGS cost effectiveness for rare and undiagnosed conditions as a diagnostic test has been published [65]. Some considerations for GS newborn screen modelling were recently published [66]. These publications are important as they highlight the complexity of carrying out an economic analysis of GS.

When compared to the individual conditions found on current newborn screening tests, sequencing-led screening *for those conditions* will likely be less sensitive given the challenges described above of balancing sensitivity and specificity in genomic variant analysis. However, when all conditions, perhaps several hundred, are considered together, genomic testing will identify many more infants with treatable conditions compared to current screening. 

A pilot programme of WGS for newborn screening will allow modelling of the potential of WES or a gene panel as a screen. This can be accomplished by examining the screening accuracy of the exome or panel data within the WGS data and comparing it to the screening accuracy of the entire WGS dataset. 

Economic analysis of newborn screening by GS is particularly complex when it overlaps with an existing screen. For countries with newborn hearing screening [67], will adding a screen by GS be economically beneficial? Because of the marginal cost of adding genes connected to hearing loss to a panel that already contains several hundred treatable disorders, the potentially small improvement in detection of infants who have or will develop hearing loss in early childhood may, nevertheless, prove worthwhile.

The Screen4Care consortium, using and validating different GS strategies, intends to demonstrate that the scalability of genetic newborn screening for rare disease will significantly reduce the cost of screening per disease and per patient compared to the current testing strategy for rare diseases which evaluates that patient with a test, one at a time.

### 2.10. Case-Finding Should Be a Continuing Process and Not a “Once and For All” Project

Developing, sustaining, and advancing screening for every newborn worldwide is an ongoing challenge [68]. In the United States, for example, both government and private funding maintain the programme in each state [69], while in many other countries the government alone is the source of funding. Advances in screening have occurred gradually with the addition of one or a few conditions at a time, allowing for integration into existing healthcare systems. The ability to derive important diagnostic information from GS beyond the initial screening result and number of disorders that can be added by GS will inform and potentially complicate further development of newborn screening.

When an approved treatment becomes available for a previously untreatable disorder adding the relevant gene to the screening test is straightforward. However, this raises the question of whether a programme should go back to previously screened cases to find additional affected individuals that may include some who are asymptomatic at the time of reanalysis. 

To build a sustainable GS newborn screening programme, an ongoing assessment evaluating health outcomes, psychosocial outcomes, costs, cost-effectiveness, and implementation is needed. This will require long-term contact with families that have been screened. Additionally, it will be important to generate data about conditions that could be included in screening in the future. Because whole genome sequencing data has the potential to detect so many disorders, creating the right consent and research governance are critical. This will result in better understanding of natural history, improved screening approaches, and optimal tracking of treatment outcomes. 

## 3. Ethical, Communication, Data Management and Sharing, Legal, and Social Implications

Newborn screening that uses sequencing requires exploration of ethical, communication, data management, legal, and social implications [70,71,72,73].

### 3.1. Ethics

A growing literature evaluating the ethical dimensions of GS as a newborn screen is developing [72,74,75]. There are broad questions, including how to choose the conditions that should be screened, what constitutes adequate and informed consent, the potential of discrimination arising from misuse of genomic data if it is stored, individual versus societal risk and benefits, and equitable access by underrepresented groups. There are also narrower questions raised by newborn GS screening, such as finding treatable, late-onset diseases [76], limiting the screen to treatable versus actionable rare diseases, and provision of heterozygous carrier status as this has preconceptional implications that allow access to prevention such as preimplantation genetic diagnosis. 

Given the potential to revisit genomic data in the future, for example, with improved screening algorithms or as new treatments become available, clear expectations need to be established for families on recontact after the initial screening result. Programmes will need to consider whether additional consent will be needed before reexamining a genome and at what age the infant, now a young person, would need to be involved in the decision to reevaluate their genome.

Research addressing these and related issues should be included in newborn screening pilot studies to help find broad consensus among all stakeholders, including policy makers and regulatory authorities.

### 3.2. Communication and Transparency

As with any screening test, the potential of both the benefits and harms of GS as a screen in pre-symptomatic newborns needs to be clearly communicated to participants. These include the potential for life-changing treatment along with diagnostic uncertainty, potential overmedicalisation, concern for genetic determinism, future unintended consequences, and the psycho-social impacts on parent-child relationships. Many efforts made to study these issues are underway [77]. 

A GS programme needs to balance the rights and needs of the child with those of the family and government agencies. Parents could choose to learn about adult genetic disorder in the child that the child when they reach adulthood would not want to know. The complexity of information and uncertainty about results from a pilot newborn GS could prompt parents to opt out of the current newborn screening.

### 3.3. Data Management

The technical aspects of data management and storage are beyond the scope of this paper. However, it is important to recognise that genomic data—particularly whole genome data—presents substantially greater challenges to store and manage than existing newborn screening data sets, particularly if offered to all babies. Standardisation of data formats and analytical approaches within and even between health systems can bring substantial benefits in terms of comparability of data and, with that, to learn from and improve screening [78].

Just as with other health data, genomic data and related clinical data need secure storage, clearly governed in line with the expectations of the parents whose newborns are screened. 

Arrangements for use beyond initial screening should be clear to families with any ongoing or future clinical or research uses supported by clear, ongoing communication and the right data access governance. This should ensure that families can have confidence that the data will only be accessed by people and organisations they expect and for purposes they expect.

## 4. The Future of Screening

With the launch of multiple research studies, we are still at the beginning of a journey, and it remains to be seen when genomics will begin to play a major role in newborn screening. GS may change our relationship with newborn screening. NGS significantly lowers the barrier to adding conditions and—with the right research consent and governance in place—has the potential to provide an engine for better evidence that will advance screening and support the search for new therapies.

## Figures and Tables

**Table 1 IJNS-08-00040-t001:** Wilson and Jungner’s principles of screening [27].

The condition sought should be an important health problem.
2. The natural history of the condition, including development from latent to declared disease, should be adequately understood.
3. There should be a recognizable latent or early symptomatic stage.
4. There should be a suitable test or examination.
5. The test should be acceptable to the population.
6. There should be an agreed policy on whom to treat as patients.
7. There should be an accepted treatment for patients with recognized disease.
8. Facilities for diagnosis and treatment should be available.
9. The cost of case-finding (including diagnosis and treatment of patients diagnosed) should be economically balanced in relation to possible expenditure on medical care as a whole.
10. Case-finding should be a continuing process and not a “once and for all” project.

**Table 2 IJNS-08-00040-t002:** Newborn genome sequencing (GS) process.

Newborn GS Process:
Sample collection (heel stick, saliva, or cord blood) and transport to laboratory
Sample accessioning in newborn GS screening laboratory
DNA extraction, quantitation, quality assessment, and plating for use in sequencing
Sequencing library preparation and quality assessment
Pooling of sequencing samples for flowcell loading and genome sequencing *
Transfer of sequence data to data analysis center
Secondary analysis at data center (mapping of reads, variant calling)
Tertiary analysis at data center (identification of variants for reporting)
Manual variant review (where necessary) and screening report generation
Transmission of final report to the physician
Physician in contact the newborn’s family

* A percentage of samples will have insufficient depth of coverage. These samples will need additional days for further sequencing to add the required coverage before the data can be sent to the data center.

## Data Availability

Not applicable.

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
