# Peer review of "Newborn Screening by Genomic Sequencing: Opportunities and Challenges"

_2409-515X, 2022, doi:10.3390/ijns8030040_

Round 1

Reviewer 1 Report

The authors have set out to address a very important potential new development in newborn screening. As they point out, the concept raises a number of important issues, some of which they suggest are different from traditional newborn screening. On the whole they are tackled well, but I think there are some issues that might benefit from further discussion or clarification.

It is true that the incremental cost of testing for more genes, is small, as mentioned in  a number of places, however, it is not only the testing that requires resources, but the follow up. It is an almost universal finding that when screening is introduced, more cases are found than amongst a non-screened population where ascertainment is based on symptomatic presentations. This difference may be because some cases are never diagnosed in life, even though they have severe manifestations. These children would probably benefit from early detection. Another reason for this difference is the inclusion of variants where the significance is uncertain and those with variable penetrance. This is a particular problem with GS. If there are many such cases, the associated costs may add significantly to the cost of the screening programme, in addition to causing unnecessary anxiety. In addition, there may need to be further confirmatory testing, which would need to be added into the costs of any false positives.

The authors wisely point out on page 5, when considering the latent stage, that what is important is not necessarily the age of symptom development, but the age at which management is best initiated.

When talking about the test, the authors use cystic fibrosis as one example with very few false positives. It is an interesting choice as one of the by-products of testing for CF is an indeterminate state known as cystic fibrosis screen positive inconclusive diagnosis. In this state it is unclear whether the baby will ever develop cystic fibrosis or a related condition. Significant numbers of parents may be condemned to a childhood of uncertainty and a lot of effort is being put into reducing this. Depending on the genes used, this can be reduced, ie increase the specificity, but it does result in a loss of sensitivity, as alluded to in a recent article in the journal.

On page 6, it is stated that in the UK 97.9% of newborns were tested. This is not quite true - 97.9% were tested and the results recorded on the child health information system (CHIS) at 17 days of age. The number tested is actually higher, ‘though they may have been tested and the results had not reached CHIS by 17 days.

The reference quoted on page 6 (no. 8) found that both sensitivity and specificity, not just sensitivity, were less with GS than analyte-base screening. This is important if wanting to reduce false positives. As the authors say, it will be important to have thought in advance, about what to do when the results from analyte testing are discrepant.

Not all programmes aim to report by seven days of age. In UK, the advice is to test on day five and so the target age for results to be available is later.

I applaud the inclusion of Table 2. Often NBS is thought to be a relatively simple process, but it is not so!

I don’t understand the sentence on page 7 that reads “As is the practice with current screening technologies, the use of a highly sensitive diagnostic test in screen-positive infants will give both physicians and families confidence to move forward with further monitoring and treatment when the GS test is positive.” Should it not be “specificity” rather than “sensitivity” as one would want to be certain that a positive result did indeed indicate a high likelihood of the condition being present. In fact, the important parameter, as these are rare diseases, would be the positive predictive value, which is highly dependent on incidence as well as specificity.

If by “true positive” is meant those who are destined to have symptomatic disease, it is true that all, by definition, will come to attention, even if not screened (page 8). However, there may be babies who are false positive in the sense that they would never have developed symptoms, in spite of their genetic make-up. These will have an impact on services as well as families, as it may only be able to ascertain their true status with the passage of time.

On page 9 is the first and, I think, only mention of carrier status. The ethical problems raised warrant a separate section of their own. Do we want to find carriers as a result of newborn screening if there is no direct benefit to the child and is removing their autonomy, in the same way as discussed on page 4?

As discussed on page 10, it will be important to be clear with families whether there is a possibility for analysing the data at a later date. Would further consent be need at that time? When would the baby, now a young person be involved in any decisions?

In their conclusion, the authors state that “GS significantly lowers the barrier to adding conditions” to screening. This may be true, but this is a two edged sword. As John Sulston said in 2002  I must also emphasise at the outset that not everything that can be done should be done.” However, I do agree that a carefully thought out evaluation is the way forward.

Author Response

REVIEWER COMMENT
It is true that the incremental cost of testing for more genes, is small, as mentioned in  a number of places, however, it is not only the testing that requires resources, but the follow up. It is an almost universal finding that when screening is introduced, more cases are found than amongst a non-screened population where ascertainment is based on symptomatic presentations. This difference may be because some cases are never diagnosed in life, even though they have severe manifestations. These children would probably benefit from early detection. 
RESPONSE
The review raises an important point about ascertainment and cost. By increasing the number of disorders sought, the cost of care must be considered.  The authors have added a sentence on line 120:  "That said, the cost of care for the additional conditions in asymptomatic newborns versus the cost in symptomatic patients identified later needs to be kept in mind."

REVIEWER COMMENT
Another reason for this difference is the inclusion of variants where the significance is uncertain and those with variable penetrance. This is a particular problem with GS. If there are many such cases, the associated costs may add significantly to the cost of the screening programme, in addition to causing unnecessary anxiety. In addition, there may need to be further confirmatory testing, which would need to be added into the costs of any false positives.
RESPONSE
The authors agree with this comment.  As noted in the section starting on line 221: "There should be a suitable test or examination." the authors emphasise the need to select variants with high specificity.  Variants of uncertain significance do not have high specificity.  The authors also point to the issue of penetrance starting on line 185: "Screening by genome sequencing may solve the cost problem when adding a disease to a screen but other challenges remain, most notably around penetrance and expressivity."

REVIEWER COMMENT
When talking about the test, the authors use cystic fibrosis as one example with very few false positives. It is an interesting choice as one of the by-products of testing for CF is an indeterminate state known as cystic fibrosis screen positive inconclusive diagnosis. In this state it is unclear whether the baby will ever develop cystic fibrosis or a related condition. Significant numbers of parents may be condemned to a childhood of uncertainty and a lot of effort is being put into reducing this. Depending on the genes used, this can be reduced, ie increase the specificity, but it does result in a loss of sensitivity, as alluded to in a recent article in the journal.
RESPONSE
It is certainly true that the current newborn screening starting with immunoreactive trypsin testing from the bloodspot card yields cases of CFSPID (cystic fibrosis screen positive, inconclusive diagnosis).  The authors specifically reference this problem (reference 52: J. Barben et al., “Updated guidance on the management of children with cystic fibrosis transmembrane conductance regula-tor-related metabolic syndrome/cystic fibrosis screen positive, inconclusive diagnosis (CRMS/CFSPID).,” J Cyst Fibros, vol. 20, no. 5, pp. 810–819, 2021, doi: 10.1016/j.jcf.2020.11.006.) 

The two paragraphs starting on line 227 discuss how increased specificity will lower sensitivity:
"In GS, when there is strong evidence that a variant or class of variant is disease causing and absent from individuals without the disease, it will have higher specificity. The need to carefully select variants with high specificity will be counterbalanced by the adverse impact that will have on sensitivity. This balance between sensitivity and specificity will differ between genes.  For example, for the genes associated with cystic fibrosis (CFTR) and spinal muscular atrophy (SMN1) where a large fraction of the causal variants is known, GS will have high sensitivity.  This is the ideal situation when screening for rare genetic diseases by GS; the test will find most of the affected newborns with very few false positives.  
Unfortunately, for many genetic disorders there are only a limited number of cases and therefore very few known pathogenic variants.  Even if these known variants are combined with presumed pathogenic variants (such as premature stop variants and frameshift variants in disorders where loss of function is the mechanism of disease), the sensitivity will be lower."

REVIEWER
On page 6, it is stated that in the UK 97.9% of newborns were tested. This is not quite true - 97.9% were tested and the results recorded on the child health information system (CHIS) at 17 days of age. The number tested is actually higher, ‘though they may have been tested and the results had not reached CHIS by 17 days.
RESPONSE
The authors have rewritten the sentence starting on line 254.  It now reads: "In the latest report from The National Health Service (NHS) newborn blood spot screening programme in the United Kingdom, for example, 97.9% of babies were tested and recorded on the Child Health Information System (CHIS) at 17 days."

REVIEWER
The reference quoted on page 6 (no. 8) found that both sensitivity and specificity, not just sensitivity, were less with GS than analyte-base screening. This is important if wanting to reduce false positives. As the authors say, it will be important to have thought in advance, about what to do when the results from analyte testing are discrepant.
RESPONSE
The authors have rewritten the sentence starting line 280. It now reads: "Additionally, the analyte-based screen now in use has greater sensitivity and specificity for the disorders currently tested than genomic testing"

REVIEWER
Not all programmes aim to report by seven days of age. In UK, the advice is to test on day five and so the target age for results to be available is later.
RESPONSE
The authors have rewritten the sentence starting on line 293.  It now reads: "At present, many screening programmes aim to report within seven days of birth"

REVIEWER
I don’t understand the sentence on page 7 that reads “As is the practice with current screening technologies, the use of a highly sensitive diagnostic test in screen-positive infants will give both physicians and families confidence to move forward with further monitoring and treatment when the GS test is positive.” Should it not be “specificity” rather than “sensitivity” as one would want to be certain that a positive result did indeed indicate a high likelihood of the condition being present. In fact, the important parameter, as these are rare diseases, would be the positive predictive value, which is highly dependent on incidence as well as specificity.
RESPONSE
The authors have rewritten the sentence starting on line 321.  It now reads "As is the practice with current screening technologies, the use of a diagnostic test with a high positive predictive value in screen-positive infants will give both physicians and families confidence to move forward with further monitoring and treatment when the GS test is positive."  It is interesting to consider the issue of incidence.  The prior probability of disease before genome testing is low as the genetics diseases have low incidence.  The probability of disease when a genome result finds pathogenic or likely pathogenic variant in an asymptomatic newborn are a great deal higher.  Therefore the posterior probability after applying a diagnostic test that is positive to a newborn with a positive molecular result will be quite high.  The molecular result will affect the PPV of a diagnostic test.

REVIEWER
If by “true positive” is meant those who are destined to have symptomatic disease, it is true that all, by definition, will come to attention, even if not screened (page 8). 
RESPONSE
The authors have rewritten the sentence starting on line 363.  It now reads "Clearly, the impact will depend on how many conditions are screened and the thresholds used for return of positive findings, noting that children with a true positive GS screen (those destined to have symptomatic disease) already have the disease and will eventually come to specialist attention."

REVIEWER
However, there may be babies who are false positive in the sense that they would never have developed symptoms, in spite of their genetic make-up. These will have an impact on services as well as families, as it may only be able to ascertain their true status with the passage of time.
RESPONSE
The authors would agree that those newborns with both a positive molecular test and a positive diagnostic test that is used in current clinical practice may have lower penetrance and milder expressivity.  This is pointed out on line 185: "Screening by genome sequencing may solve the cost problem when adding a disease to a screen but other challenges remain, most notably around penetrance and expressivity."

REVIEWER 
On page 9 is the first and, I think, only mention of carrier status. The ethical problems raised warrant a separate section of their own. Do we want to find carriers as a result of newborn screening if there is no direct benefit to the child and is removing their autonomy, in the same way as discussed on page 4?
RESPONSE
The authors would agree that carrier screening and other uses of genome sequencing data from a newborn raise many ethical issues as noted in references 72, 74 and 75 and mentioned in the ethics section starting on line 429.  Programmes such as Screen4Care are taking a different approach to some of these issues compared to the UK Newborn Genomes Programme.  Given the controversy, analysis of the many pros and cons of these issues is beyond the scope of this paper but well-covered in the references mentioned.

REVIEWER
As discussed on page 10, it will be important to be clear with families whether there is a possibility for analysing the data at a later date. Would further consent be need at that time? When would the baby, now a young person be involved in any decisions?
RESPONSE
The authors have added the sentence starting on line 446.  It now reads "Programmes will need to consider whether additional consent will be needed before reexamining a genome and at what age the infant, now a young person would need to be involved in the decision to reevaluate their genome."

REVIEWER
In their conclusion, the authors state that “GS significantly lowers the barrier to adding conditions” to screening. This may be true, but this is a two edged sword. As John Sulston said in 2002  “I must also emphasise at the outset that not everything that can be done should be done.” However, I do agree that a carefully thought out evaluation is the way forward.
RESPONSE
The authors are in full agreement.  In the 'future of screening' section, we believe that our emphasis on research and governance conveys the need for a cautious and fully transparent approach.

Reviewer 2 Report

None

Author Response

There were no Comments and Suggestions for Authors.

Reviewer 3 Report

Thank you for contributing this thoughtful commentary. Through the lens of Wilson and Jungner screening criteria, the manuscript provides a rich introduction and overall reflection of a complex NBS by GS landscape replete with opportunities and challenges. Many excellent points are raised, such as cautionary advice not to harm the important value of current NBS programs while balancing potential benefits with identification of many more disorders that may be treatable. 

From this reviewer's perspective, there are a few areas of the manuscript that may benefit from clarification as well as modifications within two WJ Criteria sections. Moreover, it would be helpful to revise formatting to improve delineation of Intro and the Ethical/Communication/etc sections from the WJ Criteria subheader paragraphs. Suggestions for consideration are raised below:

line 34: for contrast from from solely PKU, it would be helpful if some examples provided as to the current number of disorders screened in a few countries/states.

line 46-7: please clarify the distinction made between what this paper provides versus "rather than recommendations"

line 120: Please describe what is Screen4Care consortium as some readers may not know.

line 129: please explain why authors consider WJ  "appropriate" (term used by authors) or select another descriptor.

line 190: Reviewer's recommendation is to develop this section as overly brief in comparison to other sections.

line 225-275: This section is disproportionate in size (overly large) to other sections and many key points get obscured. Some ideas presented here may not precisely fall under this header.

line 293: please remove comma between test and. preferred

line 346-7: please describe why important these were just "published"

416: many efforts made

429: citation #78 not provided

Author Response

From this reviewer's perspective, there are a few areas of the manuscript that may benefit from clarification as well as modifications within two WJ Criteria sections. 

REVIEWER
Moreover, it would be helpful to revise formatting to improve delineation of Intro and the Ethical/Communication/etc sections from the WJ Criteria subheader paragraphs. 
RESPONSE
The authors have used bold type and capital letters to delineate the sections of the paper and bold type and lower case for subsections.  These changes are subject journal editorial approval.

REVIEWER
line 34: for contrast from from solely PKU, it would be helpful if some examples provided as to the current number of disorders screened in a few countries/states.
RESPONSE
The authors have add this sentence now at line 36.  "By 2015, the United States Secretary of Health and Human Services’ Advisory Committee on Heritable Disorders in Newborns and Children recommended the inclusion of 32 conditions on its Recommended Uniform Screening Panel.  These have been adopted by all states with some state to state variability [2].

REVIEWER
line 46-7: please clarify the distinction made between what this paper provides versus "rather than recommendations"
RESPONSE
The authors agree that the paper provides both issues for consideration and some suggestions.  Therefore we have rewritten the sentence now at line 49 to read: "We focus primarily on issues for consideration but provide some suggestions on how conditions might be chosen and how screening by genomic testing could be implemented."

REVIEWER
line 120: Please describe what is Screen4Care consortium as some readers may not know.
RESPONSE
The author had rewritten the sentence now starting on line 131.  It now reads: "This is one of the main rationales behind the creation of the Screen4Care consortium, a European collaborative Innovative Medicines Initiative [13] and the Newborn Genomes Programme in the United Kingdom [12]. 

REVIEWER
line 129: please explain why authors consider WJ  "appropriate" (term used by authors) or select another descriptor.
RESPONSE
The author had rewritten the sentence now starting on line 145.  "Wilson and Jungner [25] set out principles for choosing conditions that can be used for screening and reporting (Table 1)."   

REVIEWER 
line 190: Reviewer's recommendation is to develop this section as overly brief in comparison to other sections.
RESPONSE
The authors agree that the point of this screening criteria may not be clear.  At line 213 we have added this sentence: "This is a key attribute of any disorder under consideration if there is to be a timely intervention."  

REVIEWER
line 225-275: This section is disproportionate in size (overly large) to other sections and many key points get obscured. Some ideas presented here may not precisely fall under this header.
RESPONSE
The authors appreciate that this section is long.  But acceptance by parents of genome sequencing with its attendant issues will be challenging.  In each paragraph we try to describe an aspect of screening by genome sequencing that will have to be explained to families and found acceptable.

REVIEWER
line 293: please remove comma between test and. preferred
RESPONSE 
comma removed

REVIEWER
line 346-7: please describe why important these were just "published"
RESPONSE
Now at line 379 we have added this sentence: "These publications are important as they highlight the complexity of carrying out an economic analysis of GS."

REVIEWER
416: many efforts made
RESPONSE
The author had rewritten the sentence now starting on line 458: "Many efforts made to study these issues are underway [77]." 

REVIEWER
429: citation #78 not provided
RESPONSE
The citation numbering has been corrected.